# Evaluating the Efficacy of Plant Extracts in Managing the Bruchid Beetle, *Callosobruchus maculatus* (Coleoptera: Bruchidae)

**DOI:** 10.3390/insects15090691

**Published:** 2024-09-12

**Authors:** Rasheed Akbar, Brekhna Faheem, Tariq Aziz, Amjad Ali, Asmat Ullah, Imtiaz Ali Khan, Jianfan Sun

**Affiliations:** 1Institute of Environment and Ecology, School of Environment and Safety Engineering, Jiangsu University, Zhenjiang 212013, China; rasheed.akbar@uoh.edu.pk; 2Department of Entomology, Faculty of Physical and Applied Sciences, The University of Haripur, Haripur 22620, Pakistan; 3Department of Zoology, Abdul Wali Khan University Mardan, Mardan 23200, Pakistan; 4Faculty of Civil Engineering and Mechanics, Jiangsu University, Zhenjiang 212013, China; 5School of Material Science & Engineering, Jiangsu University, Zhenjiang 212013, China; 6College of Pharmaceutical Sciences, Zhejiang University of Technology, Hangzhou 310012, China; 7Department of Entomology, The University of Agriculture Peshawar, Peshawar 25130, Pakistan; 8Jiangsu Collaborative Innovation Center of Technology and Material of Water Treatment, Suzhou University of Science and Technology, Suzhou 215009, China; 9Key Laboratory of Tropical Medicinal Resource Chemistry of Ministry of Education, Hainan Normal University, Haikou 571158, China

**Keywords:** bruchid beetle, integrated pest management, repellency and toxicity

## Abstract

**Simple Summary:**

This research reveals the efficacy of five plant extracts (*Nicotiana tabacum*, *Nicotiana rustica*, *Azadirachta indica*, *Thuja orientalis*, and M*elia azedarach*) against *Callosobruchus maculatus*. The extracts were tested at six different concentrations with four replications. Phytochemical analysis revealed varying amounts of alkaloids, flavonoids, saponins, diterpenes, phytosterols, and phenols. Among the plants, *A. indica* showed the highest effectiveness, with the lowest infestation rate (16.65%), seed weight loss (7.85%), mean oviposition (84.54), and adult emergence (58.40%). Conversely, *T. orientalis* was the least effective. Toxicity analysis using probit analysis indicated that *N. tabacum*, *N. rustica*, and *A. indica* had the highest toxicity, as demonstrated by the lowest LC_50_ and LC_90_ values. Repellency tests showed that *A. indica* exhibited 100.00% highest repellency at a 3% concentration after 48 h against *C. maculatus*. This study suggests incorporating these plant extracts, especially *N. rustica*, *N. tabacum*, *A. indica*, and *M. azedarach*, into integrated pest management programs, to manage *C. maculatus*.

**Abstract:**

An estimated 2000 plant species have been employed for pest control worldwide. The use of these botanical derivatives is thought to be one of the most cost-effective and sustainable options for pest management in stored grain. The present study was designed to assess the efficacy of five plant extracts viz; *Nicotiana tabacum* L., *Nicotiana rustica* L., *Azadirachta indica* A. Juss., *Thuja orientalis* L., and *Melia azedarach* L. against *Callosobruchus maculatus* L. Plant species extracts were applied at six different concentrations, i.e., 0.5, 1.0, 1.5, 2.0, 2.5, and 3.0% in four replications. The phytochemical analyses of ethanolic extracts of five plant species showed variable amounts of phytochemicals i.e., alkaloids, flavonoids, saponins, diterpenes, phytosterol, and phenols. Total phenolic and flavonoid compounds were also observed. The efficacy of *A. indica* was highest, characterized by the lowest infestation rate (16.65%), host seed weight loss (7.85%), mean oviposition (84.54), and adult emergence (58.40%). In contrast, *T. orientalis* was found to be the least effective against *C. maculatus*, with the highest infestation rate of 25.60%, host seed weight loss of 26.73%, mean oviposition of 117.17, and adult emergence rate of 82.01%. Probit analysis was performed by estimating LC_50_ and LC_90_. The toxicity percentages of *N. tabacum* (LC_50_ = 0.69%, LC_90_ = 14.59%), *N. rustica* (LC_50_ = 0.98%, LC_90_ = 22.06%), and *A. indica* (LC_50_ = 1.09%, LC_90_ = 68.52%) were notable in terms of the lower LC_50_ and LC_90_ values after the 96-h exposure period against *C. maculatus*. Repellency was assessed by using the area preference and filter paper method. The repellency of *C. maculatus* on plant extracts increased with the increasing dose and time, such that it was the highest after 48 h. Likewise, at a 3% concentration, *A. indica* demonstrated 100.00% (Class-V) repellency followed by *N. tabacum* (96.00%, Class-V), *N. rustica* (74%, Class-IV), *M. azedarach* (70.00%, Class-IV), and *T. orientalis* (68.00%, Class-IV). Based on the findings of this study, we recommend integrating *N. rustica*, *N. tabacum*, *A. indica*, and *M. azedarach* for effective management of *C. maculatus* and highlight the potential of these plant species in the formulation of new biocidal agents.

## 1. Introduction

The issue of stored grain pest infestation has become a global concern due to its impact on both the quality and quantity of stored agricultural products [1]. Agriculture is vital to human existence, as about 90% of the world’s population utilizes and consumes agricultural products daily [2]. The health of a country’s agriculture sector directly affects that country’s progress [3]. Global agriculture aims to boost the GDP (gross domestic product) through the agricultural industry and provide enough food for the planet’s rising population [4].

Diseases and damage from insect pests are the principal factors limiting food production in developing nations [5]. An estimated 40% of the loss occurs in countries lacking adequate storage facilities. When available, pesticides are primarily utilized to control these pests. These synthetic pesticides have several downsides, i.e., they are applied indiscriminately, eliminate natural adversaries, alter the ecology, and damage the environment [6]. These unacceptable environmental health hazards demand a novel, low-cost approach to managing insect pests that would both preserve the environment and maintain pest levels below economic thresholds [7]. Cowpea (*Vigna unguiculata*) (Fab.) is a significant pulse that is primarily grown in Pakistan and affected by *C. maculatus*. Cowpeas, mung beans, and other pulses are produced and stored on a large scale with significant risk due to *C. maculatus* [8]. In semi-arid and sub-humid regions, cowpea forms an important source of food and income generation for resource-poor farmers [9].

Currently, organophosphorus, chlorinated hydrocarbons, and pyrethroids are frequently used to control storage grain insect pests in Pakistan and other developing nations; nevertheless, these chemicals may be dangerous to both human and animal health as well as the agroecosystem [10]. Furthermore, regular exposure to these synthetic chemicals disrupts biological control mechanisms, which are mostly upheld by natural enemies, and result in epidemics of minor insect pests [11]. The practice of preserving crops with chemicals produced by plants has a long history [5]. Frequently, stored commodities are protected by combining grains with plant-based protectants [12]. The practice of managing stored product pests with natural-based materials is receiving considerable attention due to its potential to minimize environmental and health-related dangers. The use of plant extracts in the treatment of insect pests is becoming more popular [13]. More research is required to explore the potential insecticidal efficiency of native plants.

This study, which evaluates the efficacy of plant extracts for managing the bruchid beetle, *C. maculatus*, is significant in the context of sustainable agriculture and pest management. Agricultural pests, such as the bruchid beetle, pose a substantial threat to food security by causing damage to stored legumes. Exploring plant extracts as potential biopesticides aligns with sustainable agriculture principles, offering environmentally friendly alternatives to synthetic chemical pesticides. This approach is in line with the global push toward sustainable and eco-friendly pest management strategies [14]. The potential success of plant extracts in controlling *C. maculatus* could contribute to the development of integrated pest management systems that minimize the ecological impacts of pest control measures [15]. Additionally, findings from this study may support the broader movement toward the development of botanical insecticides, which are considered safer for non-target organisms and the environment [16]. Furthermore, this research contributes to the validation of traditional ecological knowledge regarding the insecticidal properties of certain plants. Many communities have relied on traditional practices for pest control, and this study’s findings may provide scientific validation for such Indigenous knowledge, enhancing the integration of traditional and modern approaches to pest management [17]. By bridging the gap between traditional wisdom and scientific investigation, this study adds a cultural dimension to the significance of utilizing plant extracts for pest control, fostering a more comprehensive and inclusive approach to sustainable agriculture.

The purpose of this research is to thoroughly evaluate the insecticidal efficacy of various plant extracts against *C. maculatus*, a major pest that infests stored legumes and causes significant post-harvest losses. By identifying effective botanical alternatives to synthetic pesticides, which often pose environmental and health-related risks, this study aims to contribute to the development of sustainable pest management strategies.

## 2. Materials and Methods

### 2.1. Collection and Rearing of C. maculatus

*Callosobruchus maculatus* specimens were collected along with infested mung bean grain from various stored grain godowns in Swabi—a tehsil located in the Swabi District (34.1331° N, 72.4495° E), Razar (34.2554° N, 72.4021° E), Topi (34.1676° N, 72.6861° E), Lahore (34.1270° N, 72.4741° E), Haripur, which is a tehsil located in the Haripur District (33.9946° N, 72.9106° E), Ghazi (34°01′05″ N, 72°39′04″ E), and Khanpur (33.8020° N, 72.9077° E), in Khyber Pakhtunkhwa, Pakistan, and transported to the Entomological Laboratory of the University of Haripur (33.9781° N, 72.9128° E) for identification. The collected *C. maculatus* species were identified by the method followed in Ref. [18]. The inner surface of the hind femur is smooth, with the inner tooth usually as long as, or slightly longer than, the outer tooth. In fully developed individuals, the pronotum displays a black cuticle and is covered with golden setae, except on the basal median elevations, which extend well past the posterior edge and are covered in white, scale-like setae. The eyes are deeply notched, prominent, and bulbous. The male genitalia are distinctive, featuring a median lobe with two longitudinally sclerotized, denticulate regions near the middle, and robust, broadly spatulate parameres. Also, the coloration on the plate covers the end of the abdomen. Generally, females are larger than males. 

### 2.2. Sample Collection and Preparation of Plant Species

Samples of *N. tabacum*, *N. rustica*, *T. orientalis*, *A. indica*, and *M. azedarach* were already collected from different locations of District Haripur and Swabi and were identified by the botanist Dr. Umar Zeb, Assistant Professor in the Department of Biology, The University of Haripur, as shown in Figure 1. Voucher no. F. No. UH/Hort/2023 was dated 1 December 2023. Then we collected the samples and took them to an entomological laboratory; all the leaves and fruits were cut and thoroughly cleaned with distilled water to remove any unwanted debris. The washed leaves were then left to shade dry at room temperature. After about 15 days, when the leaves were completely dried, they were placed in an electric grinder (Philips HL7756/00 Mixer Grinder, Eindhoven, The Netherlands) to be ground into a fine powder. The powder was sieved through a mesh, size 60 (60 Mesh Sieve, Pak lab, Karachi, Pakistan), and stored at room temperature in air-tight jars to preserve the quality. Moreover, 10% ethanol extracts for each species were prepared by dissolving 20 gm of powder into 200 mL of ethanol. After this, the stock solution was further diluted and prepared at solutions of 0.50%, 1.00%, 1.50%, 2.00%, 2.50%, and 3.00% [13]. The table featuring selected plant species extracts and their parts used in the experiment are presented in Appendix A. The remaining plant specimens were kept at the Herbarium Horticulture Department, the University of Haripur.

### 2.3. Phytochemical Screening

The ethanoic extracts of five distinct plant species were subjected to biochemical testing—following the experimental protocol outlined in the current study—to investigate the different phytochemicals, such as flavonoids, alkaloids, saponins, phytosterols, diterpenes, and phenols.

Maceration: For maceration (fluid extract), whole or coarsely powdered plant drugs were kept in contact with the solvent in a stopper container for a defined period, with frequent agitation, until soluble matter was dissolved [19].

The crude extract was mixed with a few small pieces of magnesium ribbon and then concentrated hydrochloric acid was added to it drop by drop. After a few minutes, the appearance of pink or magenta-red indicated the availability of flavonoids in the sample [20]. To detect the presence of saponin, 5 mL of distilled water was added to 1 mL of extract and vortexed for 10 min. The formation of a foam column that did not disappear with the addition of HCl was deemed as positive for saponin [21]. For phenol, ethanol extracts were screened for the presence of phenol by adding three to four drops of ferric chloride solution. The emergence of a blue-black tinge suggested the presence of phenols [22]. The ethanolic extracts of test plants were treated with chloroform, followed by filtering with Whatman No. 1 filter paper (Maidstone, United Kingdom). A few drops of pure sulfuric acid were added to the solution, vortexed for a while, and left to stand for a few minutes. A yellow hue served as a sign that phytosterols were present [23]. For qualitative alkaloid determination, a few drops of Meyer’s reagent were added to 1 mL of extract. The formation of a creamy white precipitate was considered positive for the alkaloid [24]. The extracts were dissolved in water and treated with 3–4 drops of copper acetate solution. The formation of an emerald green color indicated the presence of diterpenes [25].

### 2.4. Phenolic and Flavonoid Estimation

The total phenolic and flavonoid contents of plant extracts were determined by using the Folin–Ciocalteu Spectrophotometric method, according to the method described by Ref. [19]. Samples were read on a UV-vis spectrophotometer (Waltham, MA, USA) at 650 nm. Results are expressed as catechol equivalents (µg/mg). The tables are presented in the Appendix A.

### 2.5. Impact of Plant Extracts on the Biology of C. maculatus

#### 2.5.1. Preparation of Different Ethanolic Extract Treatments

Six distinct concentrations of plant extracts, i.e., 0.50, 1.00, 1.50, 2.00, 2.50, and 3.00% were prepared by mixing 0.25, 0.50, 0.75, 1.00, 1.25, and 1.50 g of test plants powders in 50 mL of ethanol solution, adapting a *w*/*v* (weight/volume) approach. Each solution was stirred continuously for 10–15 min and passed through filtration for further purification. The filtrates were subsequently put to use in experiments [26]. The following parameters were calculated via the equations.

#### 2.5.2. Oviposition

The oviposition preference of *C. maculatus* followed the protocol from Ref. [27], with a little modification. *Vigna radiate* grains, 20 g, were treated with six concentrations of five plant species extracts. The controls, which were treated only with ethanol, were placed in glass Petri dishes (16 cm diameter.). Batches of ten pairs of freshly emerged *C. maculatus* adult beetles of the same age and size were released at the center of the test arena, and then the Petri dishes were covered immediately with muslin cloth and tightened with a rubber band to prevent the beetles from escaping and provide sufficient aeration. The setups were kept in an incubator (Percival Scientific Inc., Perry, IA, USA) at 27 ± 2 °C, 65 ± 5% RH, and a 16:8 (L:D) photoperiod. After five days, the number of eggs oviposited on the seeds was counted. The seeds were then carefully placed into their respective jars.
Oviposition = Total number of eggs laid by a female during its lifetime

#### 2.5.3. Adult Emergence Rate

Adult emergence was recorded at a regular time interval of 24 h. After 23–35 days, the adult emergence number was recorded. According to [28], the following formula was used to calculate the adult emergence rate:$$\mathrm{Adult}\;\mathrm{emergence}\;\mathrm{rate}=\frac{\mathrm{Number}\;\mathrm{of}\;\mathrm{emerged}\;\mathrm{adults}}{\mathrm{Totallaid}\;\mathrm{eggs}\;\mathrm{count}}\times 100$$

#### 2.5.4. Percent Infestation Rate

Insect damage was assessed using the count method. One hundred seeds were randomly taken from the treated plant species extracts and, using a hand lens, were observed for the presence of a hole or burrow. The insect-damaged and undamaged grain numbers were tallied [29]. The percentage of insect-damaged seeds was then calculated, as follows:$$\mathrm{Percent}\;\mathrm{Infestation}\;\mathrm{rate}=\frac{\mathrm{Total}\;\mathrm{number}\;\mathrm{of}\;\mathrm{grains}\;\mathrm{damaged}}{\mathrm{Total}\;\mathrm{number}\;\mathrm{of}\;\mathrm{grains}}\times 100$$

#### 2.5.5. Weight Loss Percentage in Mung Bean Grains

Before and after the experiments, the weight loss percentage in mung beans was obtained by weighing the seeds. The weight loss percentage was calculated using the following formula proposed in Ref. [30]:$$\mathrm{Percentage}\;\mathrm{grain}\;\mathrm{weight}\;\mathrm{loss}=\frac{\mathrm{Gi}-\mathrm{Gf}}{\mathrm{Gi}}\times 100$$

Gi = initial grain weight before infestation.

Gf = final grain weight after infestation.

#### 2.5.6. Sex Ratio

Adult beetles can be distinguished by their size and the presence of a plate at the posterior end of their abdomen. Male beetles are smaller and lack this plate. Females are bigger and have a plate at the end of their abdomen [31].

### 2.6. Contact Toxicity Assay

The efficacy of five plant extracts against *C. maculatus* was assessed following the protocol in Ref. [32]. Six concentrations of each of the five plant extracts were tested individually against *C. maculatus*. For this purpose, the original stock solution was diluted with ethanol and I ml was applied on both sides of the glass petri dish (16 cm diameter) using a micro-applicator (Hamilton^®^, PB600-1 Repeating Syringe Dispenser, (Reno, NV, USA)). Adult *C. maculatus* of a similar size and age were selected for the experiment. Six different concentrations of each plant extract, i.e., 0.50, 1.00, 1.50, 2.00, 2.50, and 3.00%, were prepared for contact toxicity, and only ethanol was served as the control treatment. A total of 280 insects were used in four replicates for each of the six concentrations of five plant extracts and the control in a completely randomized design. The Petri dishes were checked every 24 h for mortality counts throughout the 96-h exposure period. A beetle was assumed dead if showed no movement after being prodded with a camel hair brush. Mortality rates were corrected using Abbot’s formula [33].
$$\mathrm{Mortality}\;\mathrm{of}\;C.\;maculatus\;=\frac{\mathrm{Number}\;\mathrm{of}\;\mathrm{dead}\;\mathrm{insects}}{\mathrm{Total}\;\mathrm{number}\;\mathrm{of}\;\mathrm{insects}}\times 100$$

The mortality was then changed to percentage-corrected mortality by using the following equation:$$\mathrm{Corrected}\;\%\;\mathrm{Mortality}=\frac{\%\mathrm{mortality}\;\mathrm{in}\;\mathrm{treatment}\;-\;\%\mathrm{mortality}\;\mathrm{in}\;\mathrm{control}}{100-\%\mathrm{mortality}\;\mathrm{in}\;\mathrm{control}}\times 100$$

### 2.7. Repellency Assay

The repellency assay of *C. maculatus* on plant extracts was carried out following the protocol outlined in Ref. [34]. For this experiment, Whatman No. 1 filter paper, 9 cm, was used. Half of the filter paper was treated with ethanol solution as the control; for the other half, one concertation of one plant extract was applied. A similar pattern of treatment was followed for the various concentrations of five plant extracts i.e., 0.50, 1.00, 1.50, 2.00, 2.50, and 3.00%. The treated filter papers were allowed to air dry for 5 min, after which, they were placed in Petri dishes (one each). In each Petri dish, 20 adult *C. maculatus* were released to the center. These Petri dishes were then labeled, sealed, and placed in an incubator at 27 ± 2 °C in dark conditions. The number of beetles residing on the two treated surfaces within the Petri dishes was counted after 1 h, 6 h, 12 h, 24 h, 48 h, 72 h, and 96 h. The experiment was replicated four times. The repellency of *C. maculatus* was calculated using the given equation [35]:PR = [(Nc − Nt)/Nc] ×100%
where

PR = percent repellency.

Nc = number of insects in control.

Nt = number of insects in treatment.

The plant extracts were grouped into various classes based on the repellency strength it caused in *C. maculatus* (Table 1).

### 2.8. Statistical Analysis

Samples were tested for normality using the Shapiro–Wilk test [37]. As all replicates were normally distributed, differences in means were analyzed using a one-way analysis of variance with a 95% confidence interval. A Tukey HSD test was used as a post hoc test using STATISTIX8.1 [38]. SPSS version 20 software was used to conduct the above statistical analyses. LC50 (the concentration of 50% adult mortality) and LC90 (concentration of 90% adult mortality) of *C. maculatus* were determined using the Log-Probit model [39].

## 3. Results

### 3.1. Phytochemical Evaluation of Five Plant Extracts

Qualitative analysis of plant extracts suggested the occurrence of phytochemicals (Table 2). The adulticidal actions of some of these active ingredients have been explored in the literature. Among the five plant species, high to moderate levels of phytosterols, di-terpenoids, phenols, flavonoids, saponins, and alkaloids were found in *A. indica*, *N. rustica*, and *N. tabacum*. In *M. azedarach* extract, on the other hand, flavonoids, phenols, and sterols were found in moderate concentrations, while diterpenoid alkaloids and saponins were detected in traces. In *T. orientalis*, a high level of alkaloids was detected, while the levels of all other phytochemicals under investigation were low.

### 3.2. Estimation of Total Phenols and Flavonoids

Total phenol and flavonoid contents were estimated as shown in Figure 2. *N. tabacum* exhibited the highest number of total phenol and flavonoid contents.

### 3.3. Comparative Biological Parameters

#### 3.3.1. Oviposition of *C. maculatus* on Plant Ethanolic Extracts

Oviposition of *C. maculatus* decreased significantly (102.25, 94.50, 87.50, 81.75, 74.75, and 66.50) with all concentrations of *A. indica* extracts, whereas *N. rustica* caused a significant increase (130.75, 125.00, 120.25, 114.50, 109.00, and 103.50) (*df* = 20, *F* = 5.63, *p* = 0.000) (Figure 3a). The overall oviposition decreased with increasing concentrations of plant extracts. The mean oviposition of *C. maculatus* decreased significantly with *A. indica* extracts (84.54 eggs/female). With *N. rustica* extract, on the other hand, a significant maximum (117.17 eggs/female) suggests the least effect of *N. rustica* on the oviposition of *C. maculatus*. The overall mean oviposition was the highest with *N. rustica* at 117.17% and the lowest with *A. indica* at 84.54, as observed in Table 3.

#### 3.3.2. Adult Emergence of *C. maculatus* on Plant Ethanolic Extracts

From Table 3, it is clear that the overall mean percentage of adult emergence reached a maximum of 82.01% with *N. rustica* and a minimum of 58.40% with *A. indica*. The mean percentage of adult emergence of *C. maculatus* was significantly lower with *A. indica* extracts. In contrast, *N. rustica* extracts resulted in a significantly high rate of adult emergence (82.01%) (*df* = 20, *F* = 5.95, *p* < 0.01) (Figure 3b). Furthermore, adult emergence was significantly lower (70.94, 67.46.14, 63.45, 56.59, 49.83, and 42.11) with all concentrations of *A. indica* extracts, and significantly high with *N. rustica* treatments (88.56, 86.42, 84.00, 81.66, 78.45 and 72.95). In general, the adult emergence rate tended to decrease with an increase in the concentration of botanical extracts for all the plants.

#### 3.3.3. Host Infestation of *C. maculatus* on Plant Ethanolic Extracts

The results revealed a significantly lower mean host infestation rate of *C. maculatus* with *A. indica* extracts (16.65%), whereas, it was significantly higher with *N. rustica* (25.60%) (Figure 3c). Likewise, with six different concentrations of *A. indica*, the rates of host infestation were significantly lower (22.28%, 20.06%, 17.72%, 15.80%, 13.02%, and 10.99%), while *N. rustica* showed significantly higher host infestation rates with its six different concentrations (30.49%, 28.40%, 26.54%, 24.69%, 23.02%, and 20.43%) (*df* = 20, *F* = 2.67, *p* < 0.01). Moreover, the infestation rate of *C. maculatus* showed a decline as the concentration of plant species extracts increased. From Table 3, it is clear that the overall mean percentage of host infestation reached a maximum of 25.60% with *N. rustica* and a minimum of 16.65% with *A. indica*.

#### 3.3.4. Effects of Plant Ethanolic Extracts on Seed Weight Loss in Treated *C. maculatus*

*A. indica* ethanolic extract was the most effective against *C. maculatus* infestation in terms of significantly lower mean host seed weight loss (7.85%) and individual seed weight loss with six different *A. indica* concentrations (13.50%, 10.50%, 8.50%, 5.50%, 4.00 ± 0.2, and 2.50%). *N. rustica* extracts on the other hand, exhibited a significantly low efficacy against *C. maculatus* infestation as evidenced by significantly higher seed weight loss (26.73%) and (32.25%, 30.38%, 28.00%, 25.88%, 23.13%, and 20.75%) (*df* = 20, *F* = 1.58, *p* < 0.01) (Figure 3d). In addition, like other parameters, with an increase in all five plant species concentrations, host seed weight loss decreased. The overall mean percentage of host seed weight loss with *N. rustica* was higher at 26.73% and lower (7.85%) with *A. indica*, as shown in Table 3.

#### 3.3.5. Percentage of Male Emergence of *C. maculatus* on Plant Ethanolic Extracts

The results demonstrated that the percentage of male *C. maculatus* in each treatment ranged from 49.55 to 50.00% (Figure 3e), with no significant difference among the various treatments. This means that the percentage of male emergence was unaffected by any increase in the concentrations of the five plant extracts. From Table 3, one can see that the overall mean maximum percentage of male *C. maculatus* was 50.18% with *A. indica* and a minimum of 50.00% with *T. orientalis*.

#### 3.3.6. Percentage of Female Emergence of *C. maculatus* on Plant Ethanolic Extracts

Like males, no significant difference was observed in the percentage of female emergence of *C. maculatus* with five plant extracts and six different concentrations. In general, it ranged from 49.57 to 50.45 (Figure 3f). This means that the female emergence rate in *C. maculatus* remained unaffected by any successive increase in the concentration of plant extracts. From Table 3, one can see that the overall mean percentage of female *C. maculatus* was 50.08% with *M. azedarach* and 49.74% with *A. indica*.

### 3.4. Toxicity of Ethanolic Plant Species Extracts

*C. maculatus* mortality after exposure to six different concentrations of five different plant species extracts for 24 h was significantly different (Table 4). In the current study, *N. tabacum* extracts showed their highest efficacy in terms of lowest LC_50_ = 7.90% and LC_90_ = 77.84%. *T. orientalis* extracts, on the other hand, demonstrated the least effectiveness resulting in LC_50_ = 14.25% and LC_90_ = 134.24%, respectively. In Figure 4a, it can be observed that *C. maculatus* mortality was significantly higher at 53.00% with *N. tabacum*, while *T. orientalis* had the lowest mortality, with a maximum of 43.00% against *C. maculatus* (*F* = 0.84, *df* = 20, *p* = 0.6543).

*A. indica* exhibited the lowest (LC_50_ = 2.35% and LC_90_ = 32.34%) followed by *N. tabacum* (LC_50_ = 2.51% and LC_90_ = 84.57%). *T. orientalis* extracts, on the other hand, were the least effective, showing the highest (LC_50_ = 9.19% and LC_90_ = 90.78%) after 48 h. *C. maculatus* mortality was significantly higher at 67.50% with *N. tabacum* and significantly lower at 42.00% with *T. orientalis* (*F* = 1.29, *df* = 20, *p* = 0.2089) (Figure 4b).

Similarly, the mortality rate of *C. maculatus* increased significantly as the concentration of plant extracts increased after 72 h (Table 4). The maximum mortality rate of *C. maculatus* was seen at 85.00% with *N. tabacum*, and a minimum mortality was seen at 61.00% with *T. orientalis* (Figure 4c) (*F* = 0.83, *df* = 20, *p* = 0.6679)*. N. tabacum* toxicity was highest in *C. maculatus* (LC_50_ = 2.54% and LC_90_ = 14.62%). *T. orientalis* exhibited lower toxicity (LC_50_ = 4.86% and LC90 = 70.18%).

*N. tabacum* and *N. rustica* exhibited the highest effectiveness, reaching 100.00% at higher concentrations. *A. indica* and *M. azedarach* demonstrated reasonable effectiveness, peaking at 91.18%. *T. orientalis* had the lowest effectiveness, with a maximum of 82.35% mortality against *C. maculatus* (*F* = 0.31, *df* = 20, *p* = 0.9979) (Figure 4d). Regarding toxicity, *N. tabacum* exhibited the lowest LC_50_ = 0.69% and LC_90_ = 14.59%, followed by *N. rustica* (LC_50_ = 0.98% and LC_90_ = 22.60%). The highest LC_50_ = 1.44% and LC_90_ = 53.83%, as shown in Table 4.

### 3.5. Repellency of C. maculatus on Plant Ethanolic Extract

The effectiveness of ethanolic extracts from five different plant species against *C. maculatus* was investigated in the laboratory. The results showed that after a 1-h exposure period, *C. maculatus* tended to avoid the treated area. This repellency of *C. maculatus* was proportional to the increase in the concentration of plant extracts. The highest repellency was seen with *A. indica* (73.00%), whereas the lowest was recorded with *T. orientalis* (43.00%) at a 3.00% concentration (Figure 5a) (*F* = 0.07, *df* = 20, *p =* 1.00). The overall mean repellency of *C. maculatus* was significantly higher at 57.08 ± 0.33% with *A. indica*, showing Class-III repellency, and lower with *T. orientalis* (25.00%), showing Class-II repellency (*F* = 13.70, *df* = 4, *p* < 0.01) (Table 5). The repellency of *C. maculatus* further increased, reaching its maximum at 76.00% with *A. indica* and lowest at 45.54% with *T. orientalis* at a 3.00% concentration (*F* = 0.60, *df* = 20, *p* = 0.8996) (Figure 5b) after a 6-h exposure period. The overall mean repellency of *C. maculatus* was also highest with *A. indica* (62.36 ± 3.50%; Class-IV) and lowest with *T. orientalis* (27.64 ± 3.00%; Class-II) *(F* = 35.16, *df* = 4, *p* < 0.01) (Table 5). After 12 h, *A. indica* exhibited the highest repellency (80.00%); the lowest (45.00%) was exhibited by *T. orientalis* against *C. maculatus* at a 3.00% concentration, as shown in (Figure 5c) (*F* = 0.03 *df* = 20, *p =* 1.000). The overall mean repellency percentage of *C. maculatus* was also highest for *A. indica* (74.44 ± 4.10%; Class-III), while *T. orientalis* exhibited 38.97 ± 3.80% (Class-II repellency). Figure 5d shows that *A.* indica consistently exhibited maximum repellency throughout all the treatments (from 58.75% to 92.86%), while *T. orientalis* had the lowest repellency (from 31.25% to 62.62%) against *C. maculatus* after 24 h (*df* = 19, *F* = 0.05, *p* = 1.00). The overall mean repellency percentage for *A. indica* was 76.45 ± 3.80% (Class-IV repellency); for *T. orientalis*, it was 45.99 ± 3.80% (Class-III repellency), as shown in Table 5 *(F* = 10.97, *df* = 4, *p* = 0.000). Similarly, after 48 h, *A. indica* exhibited the highest repellency, increasing from 67.92% to 100% across treatments, while *T. orientalis* had the lowest repellency, ranging from 35% to 67.86% against *C. maculatus*. *N. tabacum* also showed maximum repellency, reaching 95.83% in the final treatment (Figure 5e) (*df* = 19, *F* = 0.04, *p* = 1.000). The overall mean repellency percentages were higher for *A. indica* at 82.04 ± 1.90% (Class-V repellency) and *T. orientalis* at 46.00 ± 3.80% (Class-III repellency), as shown in Table 5 (*F* = 13.75, *df =* 4, *p < 0.01*). After the 72-h exposure period, the repellency of plant species decreased with the time interval. Maximum repellency was recorded for *A. indica* (95.00%) followed by *N. tabacum* at 89.58%, and the minimum (73.00%) was recorded for *T. orientalis* at 3.00% against *C. maculatus* (Figure 5f) (*F* = 0.11, *df* = 19, *p* = 1.00). The overall mean repellency percentage for *A. indica* was 76.74 ± 3.11% (Class-IV repellency); for *T. orientalis*, it was 56.03 ± 3.90% (Class-II repellency) (*F* = 7.36, *df =* 4, *p* < 0.01). Figure 5g shows that after a 96-h exposure period among the plant species extracts, *A. indica* showed the highest repellency, reaching 81.43%, and *T. orientalis* consistently exhibited the lowest repellency, reaching 50.00% against *C. maculatus* (*F* = 0.05, *df =* 20, *p* = 1.000). The overall mean maximum percent repellency was observed for *A. indica* at 64.06 ± 1.90% (Class-III repellency), and the minimum was observed for *T. orientalis* at 37.00 ± 3.80% (Class-II repellency) against *C. maculatus*, as shown in Table 5 (*F* = 8.89, *df =* 4, *p* < 0.01).

## 4. Discussion

According to the current findings, *C. maculatus* oviposition declined significantly with all *A. indica* extracts and increased with *N. rustica*, but *C. maculatus* oviposition reduced with the rise in plant extract concentrations. These findings align with those of earlier researchers. According to [40], a mixture of green gram seeds and a petroleum-based extract of neem twigs and leaves prevented the oviposition of *C. chinensis.* Similarly, *N. tabacum* extracts greatly reduced the oviposition of *C. maculatus* on cowpea seeds [41]. This may occur as a result of the components in *M. azedarach* ethanolic extracts inhibiting bruchid oviposition by causing ovarian alterations [42]. Triterpenes have a variety of insecticidal effects, such as an ovicidal impact. According to [43], ethanolic extracts of *T. orientalis* affected *Meloidogyne incognita* eggs and juveniles. The findings of the study showed that *A. indica* extracts considerably reduced the adult appearance of *C. maculatus*, whereas *N. rustica* significantly increased it. As the concentration of plant extract increased, the emergence of *C. maculatus* as an adult decreased. The current findings are similar to [44]; the concentration of Pongamia ether extracts dramatically lowered the adult appearance of *C. chinensis* due to the extract’s volatile constituents. According to [43], volatile substances may enter the egg through its breathing pores (micropiles), which could reduce the hatching efficiency. Similar to volatile substances, non-volatile substances obstruct these pores, asphyxiating the embryo moreover, and secondary chemicals potentially contribute to the insecticidal function [45]. Another study [46] showed that the ethanolic extracts of *N. tabacum* significantly decreased or prevented the adult appearance of *C. maculatus*. The findings of this study are in line with [47], which showed a decrease in the adult emergence of *C. chinensis* from grain treated with Peganum harmala extract. The current findings of *A. indica* concentrations at all levels were considerably lower, whereas *N. rustica* concentrations were higher. As plant extract concentrations rose, *C. maculatus* host infection declined, similar to previous findings [48] on *C. chinensis* infestation for black gram seeds.

The current results showed that host seed weight loss was much higher with all tested *N. rustica* concentrations and decreased with *A*. *indica* concentrations. Higher concentrations of plant extracts caused a reduction in the seed weight loss. According to [49], plant extracts can considerably add neem, castor, karanja, and sesame oils to chickpea seeds at a rate of 4.0 to 8.0 mL/kg, significantly reducing infection and weight loss, increasing the mortality rate of *C. maculatus*, and reducing grain emergence. The reduction in beetle seed damage is concentration-dependent, attributed to higher adult mortality and less mature beetle emergence in treated grains [50]. The results of the current trials showed that plant extracts had no discernible impact on altering the sex ratio, similar to Ref. [51]. In contrast, Ref. [52] showed that females were marginally more successful (1.04:1).

During contact toxicity testing, there were noticeable differences amongst plants, which was expected due to differences in the physiological make-up of insects and the chemical composition of extracts from different plant species. Furthermore, the timing of the application affected the efficacy of the plant extracts. Numerous studies have shown variations in activity, which are consistent with the results of this study. The duration of exposure could assist in clarifying this, or it might have something to do with how well the active ingredients or compounds penetrate the body. According to our findings, all concentrations of *A. indica*, *N. tabacum*, *M. azedarach*, and *T. orientalis* significantly increased the mortality. This toxicity may be attributed to the amount of active compounds. Moreover, alteration may be caused by variations in how well each active molecule dissolves in the ethanol solvent. Due to their solubility, they offered a larger yield in percentage terms. All quantities tested for the extracts of *A. indica*, *M. azedarach*, and *N. tabacum* had a high death rate for *C. maculatus.* The results from this research are somewhat in line with the results of other studies, such as [53], which highlighted that certain botanicals are harmful to pest-storage bug insects like *C. maculatus*. The mortality rates of *C. maculatus* found in this study could be attributed to the bioactive chemicals found in the examined plant species. Even though all of the plants showed promise as insecticides, the harmful effects of the plants against *C. maculatus* differed, most likely because of the diverse phytochemical contents. Numerous phytochemicals have insecticidal properties, including tannins, flavonoids, and alkaloids [54]. *N. tabacum* contains nicotine, a naturally active leaf of tobacco that paralyzes insects by interfering with nervous systems [54]. The seeds of *M. azedarach* contain melicarpine, which controls insect growth, similar to azadirachtin. It has long been known that *M. azedarach* is extremely effective against a wide range of insect pests [55]. *T. orientalis* also showed encouraging mortality rates for *C. maculatus*. Ref. [56] reported the same results, showing that an ethanolic solution of *T. orientalis* killed Meloidogyne incognita eggs and juveniles, which are the most lethal compounds against *C. maculatus*, according to their earlier research [57]. Neem has been shown to possess compounds with repellent, antifeedant, and growth-impermissive effects, including azadirachtin, meliantriol, and salanin [58]. Prior findings by [59] support our findings, showing that Lawsonia inermis caused 90.00% adult mortality and *Euphorbia balsamiferous* resulted in 96.67%. The evaluated plant species contain secondary compounds that are efficient against the test bug. According to [60], large amounts of n-hexane extract from weeds showed the strongest cytotoxic effect. Ref. [4] provided evidence of the effects of flower extracts from *Tithonia diversifolia* on *C. maculatus* mortality. Insects’ natural physiology and behavior are disrupted by a variety of toxic substances included in botanical pesticides, which also affect how well they can feed, reproduce, lay eggs, and die [61].

The current findings are consistent with those reported by [62], who similarly asserted that *A. indica* exhibited the highest repellency in *T. castaneum*, with a falling trend over time. Our results corroborate those of a number of other studies that discovered that *A. indica* stops insects from feeding [63]. Neem extracts contain the anti-insect compounds azadirachtin and salanin [64]. *A. indica* was used to manage a number of pests that affect foliage [65]. Likewise, our results align with those of [66], who also noticed that *M. azedarach*’s repellency effect fades after a 72-h exposure period. Extensive research conducted over the past three decades indicates that *M. azedarach* has a notable repellant effect on pests found in stored products [65,66]. In the past, tobacco plants (*N. tabacum* and *N. rustica*) were thought to be natural agents of anthropocide [67]. In our study, high repellency of *C. maculatus* (82%) was recorded with *N. tabacum* at a 3% concentration and 76% with *N. rustica*. These results are in agreement with those of [68], who also observed an increase in repellency at greater concentrations of plant extracts. The authors of [69] noticed that *N. tabacum* had the greatest repellency at high concentrations against *Tribolium castaneum.* However, the current findings contrast with [70], where *Lasioderma serricorne* was highly repelled by *T. orientalis* (92%). These variations could be due to the different plant parts chosen for extraction, which contain varying percentages of chemicals [71]. Many plant extracts and essential oils contain monoterpenoids, which have insecticidal properties [72]. Monoterpenoids are often flammable and moderately lipophilic chemicals and, therefore, can swiftly enter insect bodies and influence their physiology [73]. They are useful for preventing the development of insects on stored products due to their substantial volatility and fumigant effects [74]. Over the past three decades, studies have greatly improved the understanding of biopesticides. Similarly, the current findings on the ethanolic extracts of *A. indica*, *N. rustica*, *N. tabacum*, *M. azedarach*, and *T. orientalis* suggest that they can control the population of *C. maculatus* could be options for eco-friendly pest management.

## 5. Conclusions

In the present study, among the plant species’ ethanolic extracts, *A. indica* shows promising results both in biological parameters as well as a repellency against *C. maculatus*. *N. tabacum* exhibited the highest toxicity, while *A. indica*, *N. tabacum*, *N. rustica*, and *M. azedarach* showed great potential as alternative options for pesticide use, particularly in rural tropical and subtropical regions. This research is significant for several reasons. Firstly, it promotes the use of environmentally friendly pest control methods, thereby reducing the harmful impacts associated with conventional chemical pesticides. Secondly, it addresses the critical issue of pesticide resistance in *C. maculatus* populations by providing alternative solutions that may enhance long-term pest control effectiveness. Additionally, effective management of *C. maculatus* can substantially decrease post-harvest losses, thereby improving food security, especially in regions where legumes are a dietary staple. Finally, this study fills existing research gaps by exploring the efficacy of plant extracts that have not been extensively investigated, offering new insights and expanding the options for integrated pest management (IPM) programs.

## Figures and Tables

**Figure 1 insects-15-00691-f001:**
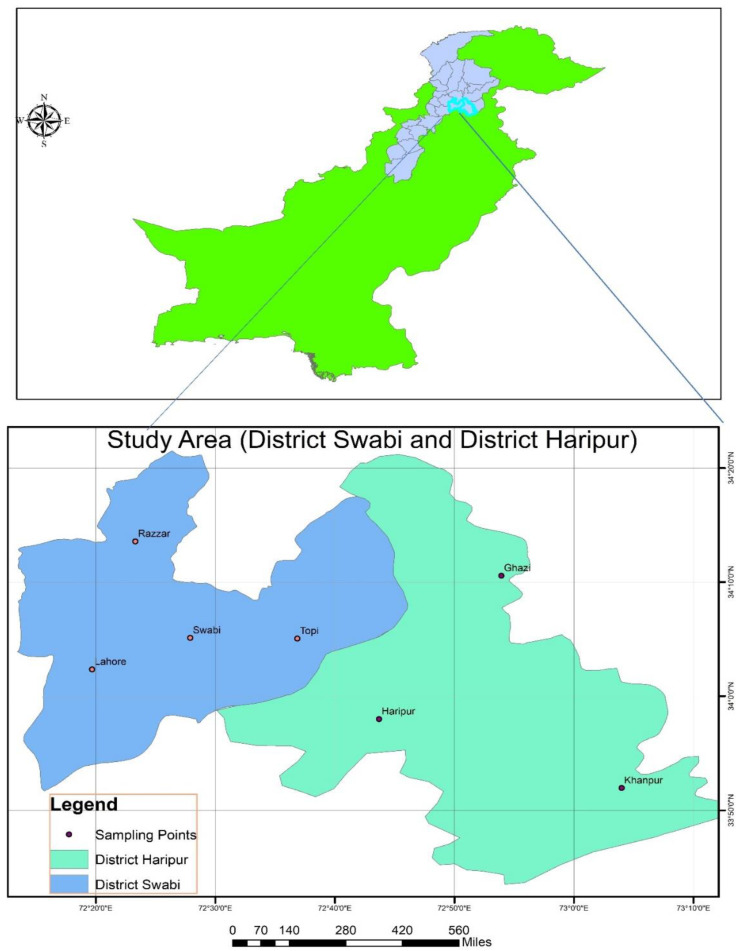
The locations from which the plant species were collected.

**Figure 2 insects-15-00691-f002:**
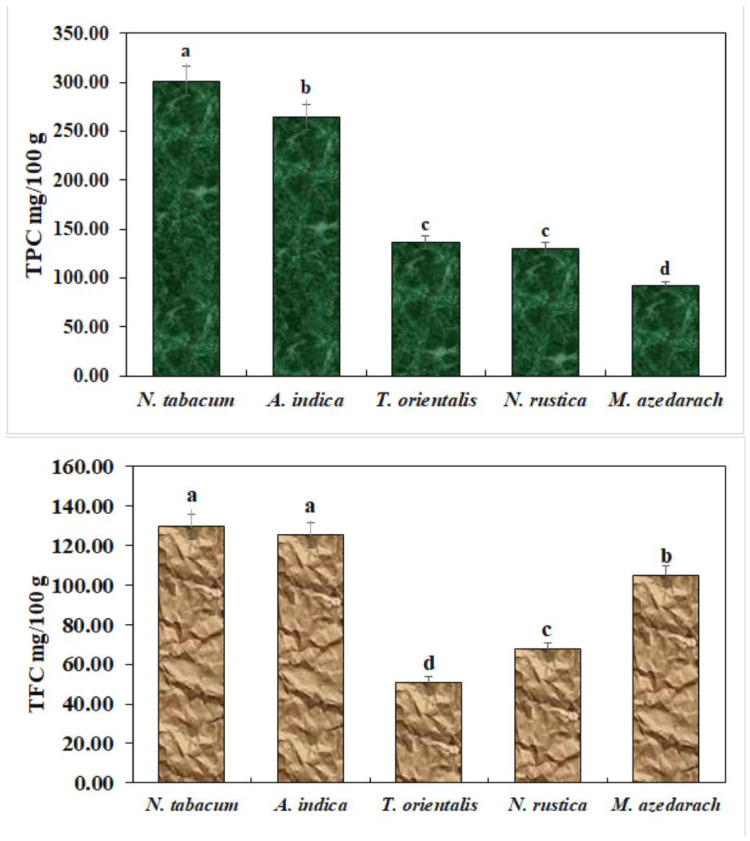
Total phenols and flavonoids in ethanolic extracts. The bars with different lowercase letters indicate that the means are significantly different from each other at *p* = 0.05.

**Figure 3 insects-15-00691-f003:**
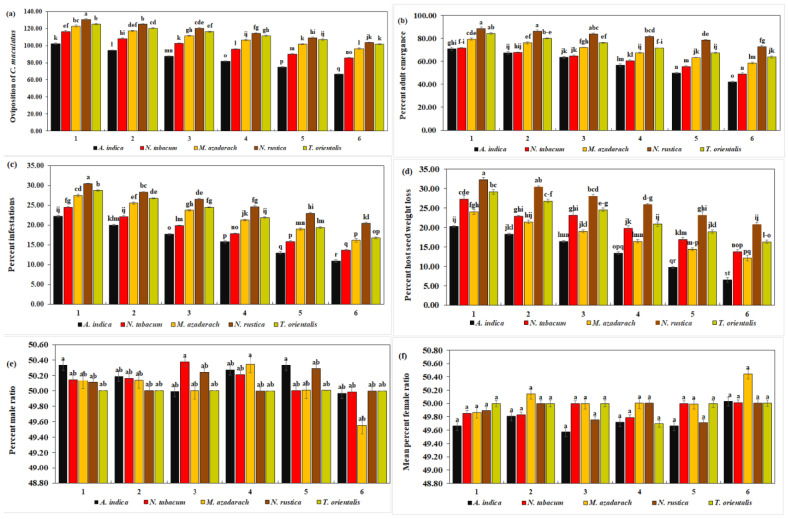
Biology of *C. maculatus*. No. of oviposition (**a**), mean % adult emergence (**b**), mean % infestation (**c**), mean % host seed weight loss (**d**), mean % male ratio (**e**), and mean % female ratio (**f**) when treated with five different ethanolic plant species extracts at six different concentrations. The bars with different lowercase letters indicate that the means are significantly different from each other at *p* = 0.05.

**Figure 4 insects-15-00691-f004:**
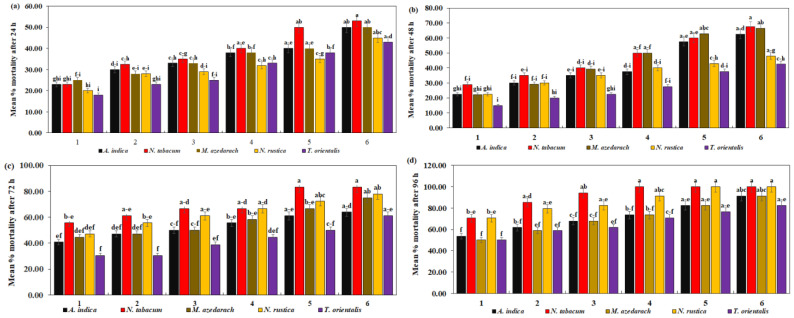
Mean percent mortality of *C. maculatus* after (**a**) 24 h, (**b**) 48 h, (**c**) 72 h, (**d**) 94 h of exposure to the ethanolic extracts of five plant species at six different concentrations. The bars with different lowercase letters indicate that the means are significantly different from each other at *p* = 0.05.

**Figure 5 insects-15-00691-f005:**
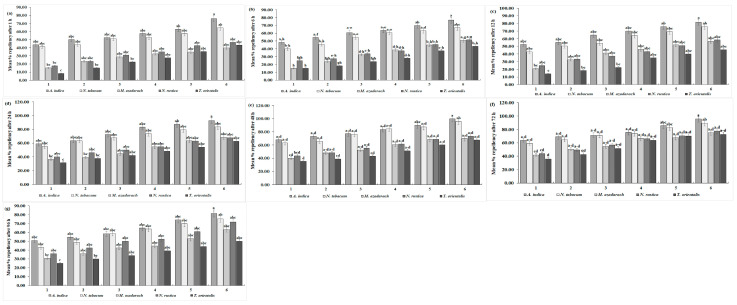
Mean percentage repellency rates of *C. maculatus* after (**a**) 1 h, (**b**) 6 h, (**c**) 12 h, (**d**) 24 h, (**e**) 48 h, (**f**) 72 h, and (**g**) 96 h of exposure to the ethanolic extracts of five plant species at six different concentrations. The bars with different lowercase letters indicate that the means are significantly different from each other at *p* = 0.05.

**Table 1 insects-15-00691-t001:** Categories of plant extracts based on the repellency strength according to [36].

Sr. No.	Class	R1
1	0	>0.01–0.1
2	I	0.1–20
3	II	20.1–40
3	III	40.1–60
4	IV	60.1–80
5	V	80.1–100

% R: Percentage of repellency rate.

**Table 2 insects-15-00691-t002:** Types of phytochemicals found in the five plant extracts.

	*N. tabacum*	*N. rustica*	*A. indica*	*T. orientalis*	*M. azedarach*
Phytosterols	+++	++	+++	+	++
Diterpenes	+++	+	++	+	+
Phenols	+++	+++	+++	+	++
Alkaloids	+++	+++	+++	+++	+
Flavonoids	++	+++	++	+	++
Saponin	++	++	++	+	+

+++ = high occurrence; ++ = moderate occurrence; + = low occurrence.

**Table 3 insects-15-00691-t003:** Overall mean no. of oviposition, % adult emergence, % infestation, % seed weight loss, and sex ratio of *C. maculatus* when mung bean is treated with five plant species extracts at six concentrations.

Plant Species	Oviposition	% Adult Emergence	% Infestation	% Seed Weight Loss	Sex-Ratio	
					% Male	% Female
*A. indica*	84.54 ± 0.57 e	58.40 ± 0.99 e	16.65 ± 0.25 e	7.85 ± 0.33 e	50.18 ± 0.35 a	49.74 ± 0.41 a
*N. tabacum*	99.79± 0.82 d	61.56 ± 0.69 d	18.99 ± 0.23 d	20.60 ± 0.72 c	50.15 ± 0.23 a	49.91 ± 0.28 a
*M. azedarach*	109.46 ± 0.78 c	69.54 ± 0.67 c	22.22 ± 0.28 c	17.88 ± 0.49 d	50.03 ± 0.21 a	50.08 ± 0.22 a
*T. orientalis*	113.79 ± 0.74 b	73.93 ± 0.59 b	23.05 ± 0.18 b	22.72 ± 0.49 b	50.00 ± 0.24 a	49.95 ± 0.29 a
*N. rustica*	117.17 ± 0.84 a	82.01± 0.90 a	25.60 ± 0.21 a	26.73 ± 0.48 a	50.11 ±0.18 a	49.89 ± 0.18 a
HSD	1.1894	1.3424	0.3774	0.8911	0.4383	0.5163
*F*	1895.04	778.42	1344.56	984.83	0.49	0.83
*p*	*p* < 0.01	*p* < 0.01	*p* < 0.01	*p* < 0.01	*p* < 0.01	*p* < 0.01

Different lowercase letters indicate that the means are significantly different from each other at *p* = 0.05.

**Table 4 insects-15-00691-t004:** LC50 and LC90 of plant extracts for *C. maculatus* after 24, 48, 72, and 96 h of exposure in the contact toxicity experiment.

Plant Species	Slope	(LC50) (g/L) (CI 95%)	(LC90) (g/L) (CI 95%)	*p*	*χ* ^2^
24 h					
*A. indica*	1.57 ± 0.12	8.94 (7.42–10.36)	55.11 (38.12–93.51)	0.763	1.85
*N. tabacum*	1.20 ± 0.11	7.90 (5.85–14.23)	77.84 (32.43–532.85	0.051	9.44
*M. azedarach*	1.20 ± 0.11	10.45 (8.51–13.97)	120.51 (67.77–278.36)	0.221	5.72
*N. rustica*	1.04 ± 0.11	9.43 (7.62–12.82)	158.62 (80.91–438.62)	0.969	0.546
*T. orientalis*	1.20 ± 0.12	14.25 (11.14–20.30)	134.24 (74.10–320.75)	0.237	5.53
48 h					
*A. indica*	1.14 ± 0.11	2.35 (2.08–2.61)	32.34 (22.40–54.21)	0.652	2.45
*N. tabacum*	0.84 ± 0.09	2.51 (2.14–2.89)	84.57 (44.61–233.80	0.402	4.029
*M. azedarach*	1.00 ± 0.10	4.36 (3.27–7.14)	82.32 (27.66–1735.86)	0.020	11.704
*N. rustica*	1.13 ± 0.10	5.02 (4.44–5.85)	67.24 (41.78–132.63)	0.370	4.276
*T. orientalis*	1.08 ± 0.12	9.19 (5.92–8.52)	90.78 (53.50–194.63)	0.500	3.355
72 h					
*A. indica*	0.85 ± 0.10	2.70 (1.61–4.03)	59.51 (18.75–5336.188)	0.002	17.13
*N. tabacum*	1.07 ± 0.10	2.54 (1.91–7.80)	14.62 (14.70–29515.52)	0.000	11.39
*M. azedarach*	1.77 ± 0.10	3.37 (2.08–3.02)	52.23 (9.95–28.63)	0.021	33.62
*N. rustica*	0.89 ± 0.09	2.84 (1.89–4.06)	76.61 (24.04–329.44)	0.071	12.08
*T. orientalis*	1.09 ± 0.11	4.86 (4.30–5.66)	70.18 (42.88–142.78)	0.609	2.69
96 h					
*A. indica*	0.85 ± 0.10	1.09 (0.71–1.41)	68.52 (34.31–227.71)	0.680	2.307
*N. tabacum*	1.07 ± 0.10	0.69 (0.008–1.39)	14.59 (6.59–2827.00)	0.000	15.89
*M. azedarach*	1.77 ± 0.10	1.39 (0.12–2.27)	39.8 (12.34–9933.500)	0.000	22.67
*N. rustica*	0.80 ± 0.09	0.98 (0.71–1.22)	22.6 (15.49–39.93)	0.293	4.941
*T. orientalis*	1.09 ± 0.11	1.44 (1.09–1.74)	53.83 (30.30–135.78)	0.239	5.50

**Table 5 insects-15-00691-t005:** Mean percentage repellency rates of *C. maculatus* after 1, 6, 12, 24, 48, 72, and 96 h of exposure to plant ethanolic extracts.

Plant Species	Mean Repellency Percentage Effect						
	1 h	6 h	12 h	24 h	48 h	72 h	96 h
*N. rustica*	32.64 ± 0.38 b	36.81 ± 0.23 b	47.62 ± 4.90 b	53.60 ± 2.80 b	58.57 ± 3.80 b	60.63 ± 4.10 bc	52.22 ± 3.40 ab
*N. tabacum*	51.81 ± 0.36 a	55.35 ± 3.30 a	69.86 ± 4.40 a	70.55 ± 3.00 a	78.95 ± 3.80 a	73.43 ± 3.30 ab	59.84 ± 3.70 a
*M. azedarach*	28.99 ± 0.20 b	34.35 ± 0.70 b	51.54 ± 3.40 b	51.07 ± 2.50 b	56.56 ± 3.80 b	59.30 ± 3.60 c	45.00 ± 3.40 ab
*A. indica*	57.08 ± 0.33 a	62.36 ± 3.50 a	74.44 ± 4.10 a	76.45 ± 3.80 a	82.04 ± 1.90 a	76.74 ± 3.11 a	64.06 ± 1.90 a
*T. orientalis*	25.34 ± 0.39 b	27.64 ± 3.00 b	38.97 ± 3.80 b	46.00 ± 3.80 b	49.36 ± 2.80 b	56.03 ± 3.90 c	37.00 ± 3.80 c
HSD	15.60	10.00	16.97	15.71	15.42	13.99	14.478
*F*	13.29	36.16	12.31	10.97	13.75	7.36	8.89
*p*	*p* < 0.01	*p* < 0.01	*p* < 0.01	*p* < 0.01	*p* < 0.01	*p* < 0.01	*p* < 0.01

Different lowercase letters indicate that the means are significantly different from each other at *p* = 0.05.

## Data Availability

All data pertinent to this work are presented in the paper. Any requests should be directed to the corresponding author.

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
