# Peer review of "Evaluating the Efficacy of Plant Extracts in Managing the Bruchid Beetle, Callosobruchus maculatus (Coleoptera: Bruchidae)"

_insects, 2024, doi:10.3390/insects15090691_

Round 1

Reviewer 1 Report

Comments and Suggestions for Authors

The authors in this study tested the efficacy of five botanicals at six different concentrations against cowpea weevil. The authors noted different levels of alkaloids, flavonoids, saponins, diterpenes, phytosterols, and phenols in the extracts of plant species. The neem extract showed high level of efficacy with reduced oviposition and adult emergence. The thuja plant exhibits reduced effectiveness against cowpea weevil. The neem extract was more repellent at high concentration. This is an interesting study but overall the write-up of article is weak and need the attention of authors. I may indicate many grammatically weak sentences, it is advised the authors should seek help from English native colleague to read the article for grammar/spell check and fluency in language.

Briefly, I have some overall comments on the article, the Introduction section need attention of authors, it should be condensed and better to be more generalized instead of focusing on a specific country. There should be fluency in the write-up and no repetition. I noticed the authors stretched much of the text, like L50-68 may be summarized in few sentences and many others. The Objectives of the study should be clearly elaborated highlighting all the parameters of the study.

The authors did not give relevant citations to some of the protocols in Methodology section, overall this section is weakly written/explained. There is need to substantially improve this section with most relevant/authentic citations of all protocols/procedures. All the required details like brand and supplier of each chemical/reagent/equipment used in this study must be provided.

The Results should completely be re-written, the authors should provide only the significant findings of the study. The results should not (L526-531) be repeated in the Conclusion section, this section should be more be generalized and future needs and bottlenecks of the study be provided here.

I am of the opinion that in the current form this article can not be accepted for its publication in this high standard journal but keeping the importance of data in view the Editor may allow authors to substantially improve the article and re-submit for consideration.

L15, replace “Callosbruchus” with “Callosobruchus”, pl check the spellings throughout ms

L28-29, provide full taxonomy of each scientific name on its first use in ms

L41, 43, italicize all the scientific names throughout ms

L44, replace “(class: IV)Base” with “(class: IV). Base”

L90, all the citations should be numbered following the style of journal

L114, start each sentence/paragraph with complete genus name of each scientific name in ms

L114, how many insects were collected from the depots?

L114, what does mean the word “depots”, although I understand but it should be replaced with appropriate word

L114, from how many depots the specimens were collected and provide their exact coordinates?

L116, provide exact coordinate of “University of Haripur”?

L116, how the specimens were identified?

L117, how the insects were cultured, give brief detail of procedure and the food provided in containers?

L117, how many male/female were released to start the culture in containers?

L122, how many locations were visited to collect samples and provide exact coordinates of each location?

L124, I can not see Figure 1 in the article, pl check

L128, give details of “electric grinder” like its brand, company and its location etc.

L129 all the details about “sieve” should be given, pl provide the brand, company and its location, same details of all equipment and chemicals/reagent used in this study should be given throughout ms

L134, I can not see supplementary file s1, pl check?

L121-131, give relevant citation to the protocol for the sample preparation

L141-171, it is better to give more details of each protocol for easy understanding of readers

L180, how long the females were survived and for how many days the data was noted?

L180-190, give citation to each of the equation

L191-193, give citation to this protocol

L202, how many times the experiment was repeated, same information for each experiment in ms should be given?

L207, give citation to this equation

L220, better to give citation to the equation

L229, Table 1, what is “&R1” and strength of repellent action was calculated?

Table 1, what does mean “1%R: Percentage of repellency rate”, should Table 1 be the part of Methodology section, it looks Table 1 is derived from McDonald et al. (1970) but I do not understand the relevancy of this table with the study?

L233, either the authors normalize the data before analysis?, if it is not the case then it is important to first normalize and then re-analyze the data. Otherwise, the details for normalization with relevant references should be given here

L235, give citation for “Tukey’s test”

L396, what does mean “Figure 5 (g)”?

Table 3, the statistics (F, df and p value) and standard error (± SE) of means should be given? The row for A. indica should not be in bold and underlined

Table 5, the statistics (F, df and p value) and standard error of means should be given? The mean values should be in two decimals in all the Tables in ms

L236, why authors use two different statistical software to analyze the data?

Figure 2, 3, 4, 5, the authors should put significance letters on the bars, the statistics (F, df and p value) should be provided in the Results section

The References should be strictly formatted following the authors guidelines of the journal

Comments on the Quality of English Language

The write-up of article is weak and need the attention of authors. I may indicate many grammatically weak sentences, it is advised the authors should seek help from English native colleague to read the article for grammar/spell check and fluency in language.

Author Response

Respected  reviewer Thank you very much for your constructive comments. We have carefully considered each of your suggestions and have addressed them point by point in our revised manuscript. The responses to each of your comments are detailed below, and the corresponding changes have been incorporated into the revised manuscript as suggested. We greatly appreciate your efforts to improve the quality of our work. Please find the attached word file for further process

Reviewer 2 Report

Comments and Suggestions for Authors

The current manuscript (Evaluating efficacy of plant extracts for the management of  Bruchid beetle Callosobruchus maculatus (Bruchidae: Coleop- tera)) investigated the insecticidal activity of 5 Eos against C. maculatus. The manuscript has two main issues:

1.      The materials and methods need to be clear specially the assays against the insect contact toxicity

2.      The results need to re-write to be readable and need to be concentrated

Another some points need to be clarifying from the authors:

Collection and rearing of insects: please add C. macullatus

2.2. Sample collection and preparation: of what?

Phytochemical screening from 1 to 6 can compacted in one paragraph.

2.5.1. Preparation of different Ethanolic extracts treatments:

The following pa-178 rameters were calculated via the equations: what the authors did to calculate this parameters?

Contact toxicity assay: what the nature of the used petri dish, plastic or glass? The volume of each concentration that spread on the petri dish?

In  line 201: i.e. 0.5, 1.50, 2.50, 3.00, and 1.50 % is wrong , please correct.

In results:

Oviposition of C. maculatus on plant ethanolic extracts: what was the assay that authors did to measure this? Also table 3

In table 3, % seed weight loss: which seeds?

Host infestation of C. maculatus on plant ethanolic extracts: which host?

3.3.4. Effect of plant ethanolic extracts treated C. maculatus on seed weight loss: where was the test of this?

The units should be fixed in all the manuscript, the authors applied concentrations in term of % then calculate the LC50 in term of g/L

Toxicity of plant ethanolic plant species extracts: the authors described in over details, the results should be in comparable with all Eos in differences only. The authors mentioned all concentrations and all mortality %. It is so much details

3.5. Repellency of C. maculatus on plant ethanolic extract: also, so much details

Discussion needs to be concentrated about the used Eos

Conclusions are so long, and need to be concluded.

Author Response

Dear Editor & reviewer, thank you so much for your valuable comments, we have addressed the comments point by point and appended response to each comment which are attached below. The changes suggested by editor/reviewers have been written in the revised manuscript.

Round 2

Reviewer 1 Report

Comments and Suggestions for Authors

I have gone through in detail the revised version of article and found that the authors substantially improved the manuscript following most of my suggestions, but there is an ample room to further improve the article. In my opinion the authors should carefully respond to each of my comment, once updated this manuscript may be accepted for it publication in Insects but after making necessary/suggested following changes in the revised version of article. It is suggested the language and spell check should be done before its publication. 

Some of the comments not responded by the authors are as follows: 

- I noticed that the significance letters on Figure 3, 4 and 5 are not legible; all the Figures should be revised and the letters should be easily readable. I did not see - - Figure 1 and there is no Supplementary file (s1) in the article

- The citation to the Tukey test should be given in the article.

- P = 0.0000 should be replaced with P < 0.01 throughout revised version of article 

It is better the authors should provide a response to each of my query in a separate letter, point to point response to reviewers comments

Comments on the Quality of English Language

It is suggested the language and spell check should be done before its publication.

Author Response

Respected Review please checked the manuscript and response sheath in word file 

Reviewer 2 Report

Comments and Suggestions for Authors

The authors replied to all comments. 

There are some minor edits. For example, the abbreviation of hours, I think is h not hr. Also, LC50 and LC90 in Table 4, need to be subscript. The modified parts in M&Ms and results need be in the same font and size of writing 

Author Response

Respected Reviewer we have addressed all the suggestion . please check the word file (Response sheath attached)
